# Effects of Ginger Intake on Chemotherapy-Induced Nausea and Vomiting: A Systematic Review of Randomized Clinical Trials

**DOI:** 10.3390/nu14234982

**Published:** 2022-11-23

**Authors:** Jihee Choi, Jounghee Lee, Kijoon Kim, Hyo-Kyoung Choi, Se-A Lee, Hae-Jeung Lee

**Affiliations:** 1Department of Food and Nutrition, College of Bionanotechnology, Gachon University, Seongnam-si 13120, Republic of Korea; 2Institute for Aging and Clinical Nutrition Research, Gachon University, Seongnam-si 13120, Republic of Korea; 3Department of Food and Nutrition, Kunsan National University, Gunsan 54150, Republic of Korea; 4Department of Food and Nutrition, Sookmyung Women’s University, Seoul 04310, Republic of Korea; 5BOM Institute of Nutrition and Natural Medicine, Seoul 05554, Republic of Korea; 6Korea Food Research Institute, 245, Nongsaengmyeong-ro, Iseo-myeon, Wanju-gun 55365, Republic of Korea

**Keywords:** ginger, chemotherapy, nausea, vomiting, systematic review

## Abstract

Nausea and vomiting are the most common side effects of chemotherapy. They must be managed because they can increase the risk of malnutrition in patients, which can adversely affect treatment. The objective of this study was to evaluate the effect of ginger supplementation as an adjuvant treatment for alleviating chemo We checked. therapy-induced nausea and vomiting (CINV). This study searched for randomized controlled trials (RCTs) related to ginger supplement intake for CINV in three electronic databases (i.e., Medline (PubMed), Embase, and Web of Science). The search period ranged from each database’s first date of service to 5 November 2021. Two investigators independently performed abstract screenings, full-text screenings, data extraction, and risk of bias analyses (ROB). The Cochrane ROB tool was used for the assessment of ROB. This study systematically reviewed 23 RCTs. The effects of ginger supplementation were compared to those of placebo or antiemetic agents. This study conducted a meta-analysis after classifying the effects of ginger supplementation on acute and delayed CINV into subgroups due to the clinical heterogeneity between these RCTs. The results showed that the incidence of acute nausea (*p* = 0.53), the incidence of delayed nausea (*p* = 0.31), the incidence of acute vomiting (*p* = 0.09), and the incidence of delayed vomiting (*p* = 0.89) were not significantly different between the ginger supplement intake group and the control group. However, it was found that the ginger supplement intake group, which took not more than 1 g of ginger supplementation per day for above four days, had significantly less acute vomiting than the control group (OR 0.30; 95% CI 0.12 to 0.79; *p* = 0.02; I2 = 36%). Ginger supplementation may reduce the incidence of acute chemotherapy-induced vomiting. However, this study could not confirm the effects of ginger supplementation on the incidence of chemotherapy-induced nausea and delayed vomiting. Therefore, it will be necessary to conduct additional studies with sufficient sample sizes using high-quality RCTs to evaluate the effects of ginger supplementations based on the results of this study.

## 1. Introduction

Chemotherapy, along with surgery and radiation therapy, is one of the most common and effective cancer treatment methods [1,2]. It prohibits the growth of cancer cells, which spread throughout the body or kill them by using drugs. However, it may cause side effects because it can damage healthy cells as well as cancer cells [3]. Most of the patients who received chemotherapy (88%) experienced one or more side effects: fatigue (80%) was the most common side effect, followed by nausea and vomiting (48%) and pain (48%) [4]. Especially, it is necessary to pay more attention to nausea and vomiting because they can deteriorate the quality of life of the patient, negatively affect food intake [5], and increase the risk of malnutrition during treatment. This is because continuous persistent malnutrition can adversely affect treatment by lowering the patient’s immunity [6]. Prescribing an antiemetic drug is one way, but only approximately 26% of patients who were prescribed antiemetics experienced that they were effective in mitigating nausea and vomiting symptoms [7]. Additionally, taking them may cause side effects such as headache and sedation [8]. Consequently, many researchers have actively conducted studies on the effects of adjunct therapies using natural products that are effective for patients experiencing chemotherapy-induced nausea and vomiting (CINV) symptoms, with only a few side effects. Ginger (*Zingiber Officinale Roscoe*) is one of the most commonly used adjunct treatments for patients with CINV. 

Ginger is originally from tropical Asia. It is a perennial crop, and it is widely used as a spice all over the world owing to the pungent and spicy taste of the rootstalk. Ginger is rich in a variety of components including phenolic compounds, polysaccharides, terpenes, lipids, and organic acids [9]. The main functional substances of ginger are phenolic compounds such as gingerols and shogaols [9]. They have various biological properties that are antioxidant, anti-inflammatory, antimicrobial, anticancer, neuroprotective, antidepressant, and antiemetic [9]. Due to these biological activities of ginger, it has been used as a traditional remedy for nausea, vomiting, indigestion, motion sickness, and morning sickness in various cultures for a long time [9,10]. It is known that ginger mitigates nausea and vomiting due to the inhibitory effects of gingerols or shogaols against 5-Hydroxytryptamine type 3 (5-HT3) receptors [11]. Gingerols and shogaols are presumed to have antiemetic effects by binding to the serotonin binding site through acting on the 5-HT3 receptor ion–channel complex [12]. 

A systematic review and meta-analysis study published in 2019 confirmed, after examining 18 parallel and crossover intervention trials, that ginger supplementation and CINV were not significantly associated [13]. However, the review study suffered from low reliability because studies included in the review were clinically heterogeneous due to different cancer types, chemotherapy, and antiemetics [13]. Therefore, there is a need to review recently updated publications. The objective of this study was to evaluate whether the ginger supplement was effective in reducing CINV compared to the use of a placebo or standard antiemetic medication in adult cancer patients receiving chemotherapy.

## 2. Materials and Methods

### 2.1. Data Sources and Searches

We conducted the literature search for randomized controlled trials (RCTs) related to ginger intake and chemotherapy-induced nausea and vomiting in 3 databases: Medline, Embase, and Web of Science (from inception to 5 November 2021; search on 5 November 2021). The complete search strategy is demonstrated in Appendix A.

### 2.2. Study Eligibility Criteria

We included human intervention trials with ginger intake from dietary supplements or foods. To be eligible, studies must have compared the effect of ginger intake alone with that of a placebo or another active comparator. Additionally, studies must have information related to chemotherapy-induced nausea and vomiting.

### 2.3. Study Selection Process

Two investigators independently screened the abstract of all citations according to study eligibility criteria. To conduct the abstract screening, the researchers separately used the Rayyan program, the open source online software. Then, two investigators independently conducted the full-text screening according to the study eligibility criteria. We resolved the conflicts by our research group consensus. The study selection is presented in Figure 1.

### 2.4. Data Extraction and Study Quality Assessment

The research conducted data extraction (e.g., study characteristics, odds ratio) of all the included studies by employing standardized data extraction forms. To examine the risk of bias (ROB) for each study, we employed the Cochrane ROB tool for randomized controlled trials. The Cochrane ROB tool consisted of five domains of bias including: (1) the randomization process; (2) deviations from the intended intervention; (3) missing outcome data; (4) the measurement of the outcome; and (5) the selection of the reported result. The researcher evaluated the ROB by selecting one of three options: low, high, or unclear risk. We conducted data reporting in accordance with the preferred reporting items for systematic reviews and meta-analyses (PRISMA) reporting guidelines.

### 2.5. Quantitative Synthesis

Due to large methodological heterogeneity (e.g., different levels of ginger intake), we conducted a random-effects meta-analysis when there were at least two unique studies that stated sufficient quantitative information for the same outcome. To quantify the level of statistical heterogeneity, we conducted the Tau-square test, the Chi-square test, and the I-square test. We conducted all calculations and meta-analyses using the Cochrane’s Review Manager software. We considered values of less than 0.5 for two-tailed *p* values as statistically significant.

## 3. Results

We identified 267 citations by the initial search. After excluding dual abstracts, we produced 159 abstracts. After screening out 135 abstracts owing to the exclusion criteria, we identified 24 articles for full-text screening. Then, we finally included 23 abstracts for this systematic review. The flow chart of the study selection process is presented in Figure 1. The study characteristics of the included RCTs are stated in Table 1. The study risk of bias assessment for the included RCTs is in Table 2.

### 3.1. Ginger Intake and Chemotherapy-Induced Nausea

#### 3.1.1. Chemotherapy-Induced Acute Nausea

The meta-analysis result of five RCTs showed that the ginger intake group showed a tendency to slightly lower the incidence of acute nausea, but there was no significant difference (Odds Ratio [OR] = 0.82, 95% CI = 0.44 to 1.52, *p* = 0.53, I^2^ = 58%) (Figure 2).

#### 3.1.2. Chemotherapy-Induced Delayed Nausea

The meta-analysis result of six RCTs presented that the ginger group showed a tendency to marginally lower incidence of delayed nausea, there was no significant difference between the ginger and the control group (OR = 0.81, 95% CI = 0.53 to 1.22, *p* = 0.31, I^2^ = 10%) (Figure 3).

### 3.2. Ginger Intake and Chemotherapy-Induced Vomiting

#### 3.2.1. Chemotherapy-Induced Acute Vomiting

The meta-analysis results of six RCTs presented that the ginger intake group showed a tendency to slightly lower the overall incidence of acute vomiting, but there was no significant difference (OR = 0.59, 95% CI = 0.33 to 1.09, *p* = 0.09, I2 = 52%). However, the odds ratio of acute vomiting was significantly decreased in subjects undertaking chemotherapy by 70% with ginger supplementation ≤ 1 g/day for > 4 days compared with the control groups (OR = 0.30, 95% CI = 0.12 to0.79, *p* = 0.02, I2 = 36%) (Figure 4).

#### 3.2.2. Chemotherapy-Induced Delayed Vomiting

The meta-analysis results of six RCTs showed a non-significant difference of the effect of ginger on the onset of delayed vomiting (OR = 0.81, 95% CI = 0.39 to 1.69, *p* = 0.89, I^2^ = 71%) (Figure 5).

### 3.3. Reported Adverse Events after Intake of Ginger

Seven studies reported adverse events, including mild gastrointestinal symptoms, constipation, reflux, heartburn, bruising, flushing, rash, fever, fatigue, diarrhea, anemia, etc. [17,20,25,29,32,35,36] (Table 3). The incidence of these adverse events was similar between the ginger group and the placebo/control group, with no significant difference in adverse effects compared to the placebo. The distribution of serious adverse events between the ginger group and the placebo/control group was similar, with fewer events in the ginger group [17]. Nine studies reported no adverse events occurred during the study [15,16,18,22,23,24,30,33,34]. Seven studies reported no information regarding adverse events [14,19,21,26,27,28,31].

## 4. Discussion

This is the most recent study that systematically reviewed and conducted a meta-analysis on randomized controlled trial (RCT) studies, which evaluated the effectiveness of ginger supplements in reducing CINV compared to placebo and standard antiemetic medication in adult cancer patients receiving chemotherapy. The results of previous meta-analysis studies showed that taking at least 1 g of ginger supplements per day for three days or more reduced the chance of acute vomiting caused by chemotherapy by 60% [13]. However, when this review study carried out a meta-analysis of the six RCTs, ginger supplements did significantly decrease the possibility of acute chemotherapy-induced vomiting, as similar results were shown in the previous systematic review. The results of this study agreed with those of previous studies [1,13,37], which revealed that ginger supplements did not significantly decrease the occurrence and severity of chemotherapy-induced nausea, the occurrence and severity of delayed nausea, and the occurrence and severity of delayed vomiting.

Although the certainty of the effect size estimated in this review was very low, it was believed that ginger supplements were effective in reducing the possibility of acute vomiting. However, the results are not conclusive because the clinician heterogeneity of the current evidence is high. Therefore, all results must be interpreted with caution until additional results are published from a study based on a solid experimental design and well-controlled samples.

Acute CINV, peaking at 5–6 h after chemotherapy, is related to 5-HT in the central and gastrointestinal tract, whereas delayed CINV, peaking at 72 h after chemotherapy, is mediated by SP in the central [38]. Ginger is rich in bioactive polyphenolic compounds, and 6-gingerol and 6-shogaol are the representative ingredients in gingerols that contribute to antiemetic actions against CINV [39]. The mechanism for the antiemetic action of ginger has not been fully understood yet. However, several mechanisms of the effect of ginger against CINV are proposed with the interactions of neurotransmitters in the central and peripheral, such as 5-hydroxytryptamine (5-HT), substance P (SP), and dopamine (DA), and the modulation of gastrointestinal motility [12,40,41,42].

Most of the monoamine neurotransmitter, 5-HT, is produced in the intestinal EC cells [43]. Chemotherapy stimulates EC cells to release 5-HT, and then activates 5-HT3R, resulting in nausea or vomiting [44]. Gingerols and its ingredients, especially 6-shogaol, significantly mitigate CINV by reducing 5-HT and blocking 5-HT3R expression [12]. Another mechanism is through mediating the substance P (SP) signaling pathway. The binding of SP to the neurokinin-1 receptor (NK-1R) results in vomiting [45]. SP is derived from substance P-precursor, preprotachykinin A (PPT-A) and Neprilysin (NEP) is an ectoenzyme that degrades tachykinins, such as SP [46]. The gingerols in ginger significantly ameliorated vomiting by reducing the SP level through decreasing of PPTA and increasing of NEP [41]. 

In addition to 5-HT and SP systems, the activation of the dopamine (DA) signaling pathway also leads to CINV. Tyrosine hydroxylase (TH) is the rate-limiting enzyme in DA synthesis, and DA activates D2-like dopamine receptors (D2R) through the dopamine transporter (DAT) resulting in an emetic response [47]. The effect of gingerols against CINV is partially due to the inhibition of DA synthesis and D2R activation caused by increasing TH and reducing DAT [47,48]. Chemotherapy may affect gastrointestinal motility, as well as that of neurotransmitters. It was reported that the delayed gastric emptying caused by chemotherapy may also be an important factor in explaining CINV [48,49,50]. The gingerols in ginger markedly improved delayed gastric emptying induced by cisplatin in a dose-dependent manner [42]. Despite several proposed mechanisms, further research is warranted to elucidate the underlying mechanism of the antiemetic action of ginger against CINV.

The studies reviewed in this study showed that the intake of ginger supplements varied from 160 mg/day to 15 g/day. It was also found that various intake methods were used: tea, capsulated ginger powder, taking the ginger powder with other foods such as yogurt, and capsulated ginger extract. Therefore, it is highly uncertain what is the most ideal dosage and method of taking ginger supplements for effectively reducing CINV because studies the clinical heterogeneity between studies is fairly high. It is believed that taking 1g or more of ginger supplements for 3 days or longer is potentially helpful in reducing the occurrence of acute vomiting.

Ginger is a spice that has been consumed by mankind for more than thousands of years [38]. Since people consume small amounts of it in daily life from cooked dishes, it does not require much caution. However, when consuming it in the form of a supplement, people may consume excessive amounts. Therefore, it is necessary to pay caution when consuming ginger supplements. Some randomized double-blind trials argued that ginger supplements were a non-pharmacological treatment that could reduce nausea and vomiting during pregnancy and its efficacy and safety were proven even for pregnant women [51,52,53,54]. However, these results should be interpreted with caution because the amount of intake varied between studies and the duration of intake was different between them. As a result, countries such as Finland prohibit pregnant women from taking ginger supplements due to the lack of solid scientific evidence for the benefits of ginger [55]. Moreover, the cytotoxicity [56,57,58] and mutagenic effects [59,60,61] of gingerols, the main functional substances of ginger, were tested in in vitro studies. Therefore, it is difficult to extrapolate these results to humans. However, these can be grounds that more research is needed to understand the caution against consuming ginger supplements that contain high levels of gingerol extract, which is sometimes pursued rather than consuming ginger in the diet and ensure the safety of ginger supplements and caution against an overdose of ginger supplements that contain high doses of gingerol extract, rather than consuming ginger in the diet.

This is the most recent systematic review and meta-analysis study on RCTs to determine whether ginger supplements were effective in reducing CINV compared to placebos or standard antiemetic medication. Conclusively, it was found that ginger supplementation was effective in reducing the possibility of acute vomiting. However, since the subjects of the reviewed studies took ginger supplements, it may be difficult to observe the same effect by simply adding ginger to the daily diet. Although several studies were identified in this systematic review, substantial clinical heterogeneity was reported through a lack of reporting or large variations across studies in ginger interventions; the sources of clinical heterogeneity were varying CTx and antiemetic agents, as well as the type of cancer and groups of patients. It is challenging to draw reliable conclusions due to the large clinical heterogeneity between studies. Therefore, further studies are needed to gain enough confidence regarding the effects of ginger on CINV-related outcomes.

## Figures and Tables

**Figure 1 nutrients-14-04982-f001:**
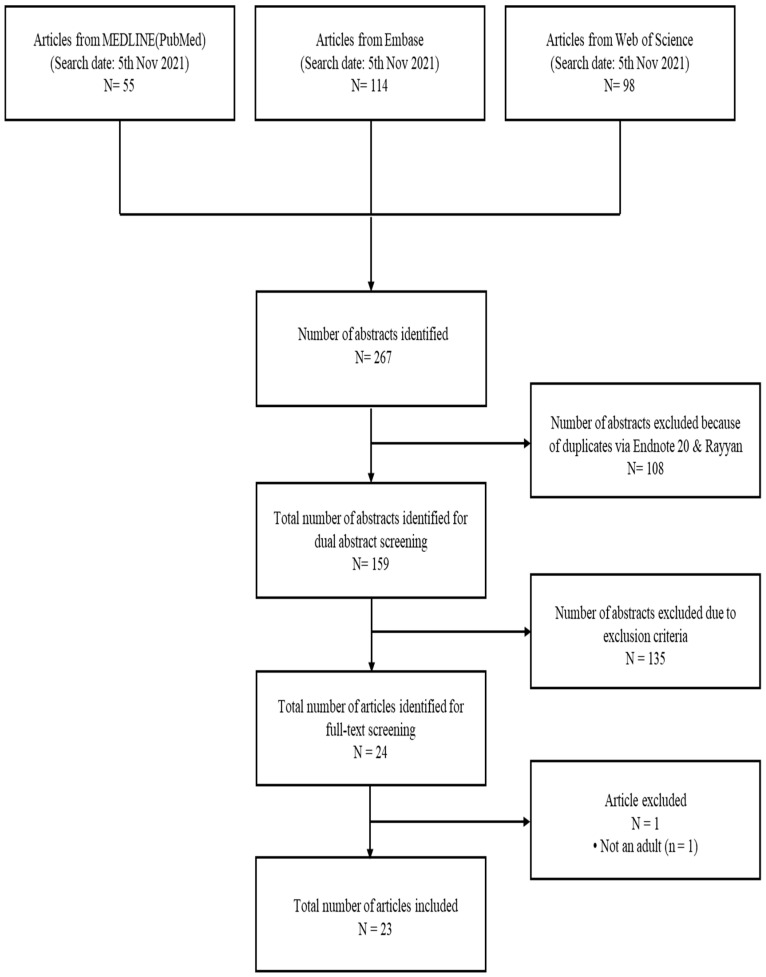
Literature search and study selection process.

**Figure 2 nutrients-14-04982-f002:**
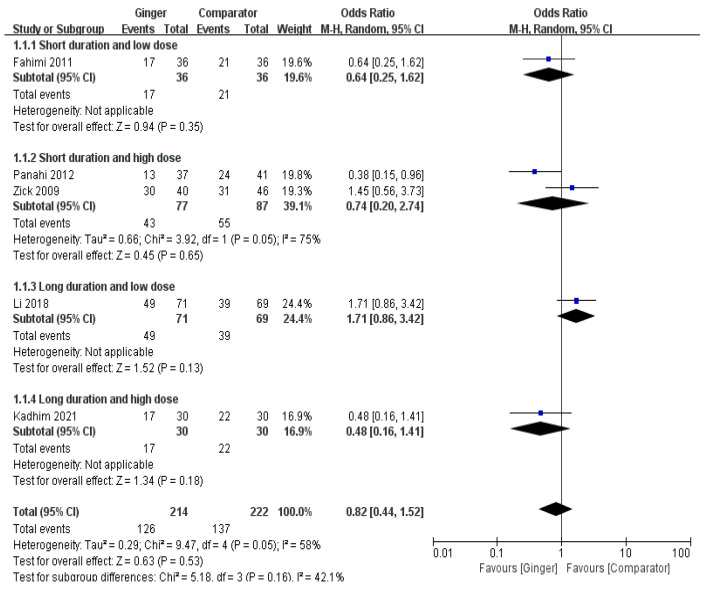
Effects of ginger intake on chemotherapy-induced acute nausea. Short duration ≤ 4 days; Long duration > 4 days; Low dose ≤ 1 g/day; High dose > 1 g/day. Fahimi 2011 [19], Panahi 2012 [28], Li 2018 [23], Kadhim 2021 [21].

**Figure 3 nutrients-14-04982-f003:**
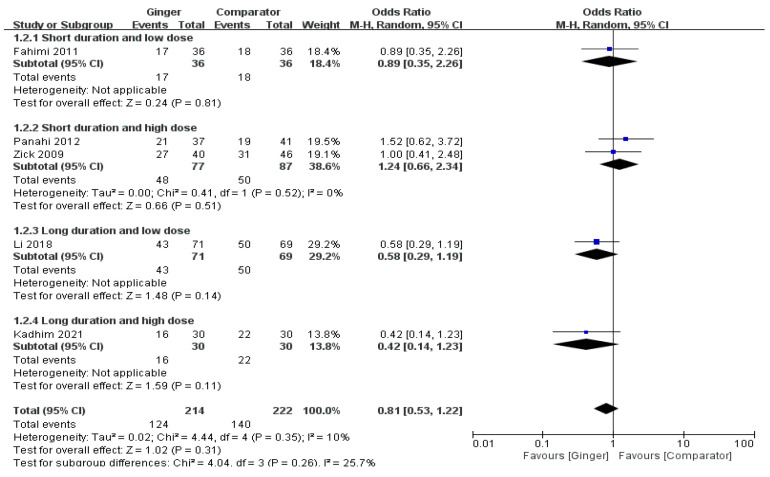
Effects of ginger intake on chemotherapy-induced delayed nausea. Short duration ≤ 4 days; Long duration > 4 days; Low dose ≤ 1 g/day; High dose > 1 g/day. Fahimi 2011 [19], Panahi 2012 [28], Li 2018 [23], Kadhim 2021 [21].

**Figure 4 nutrients-14-04982-f004:**
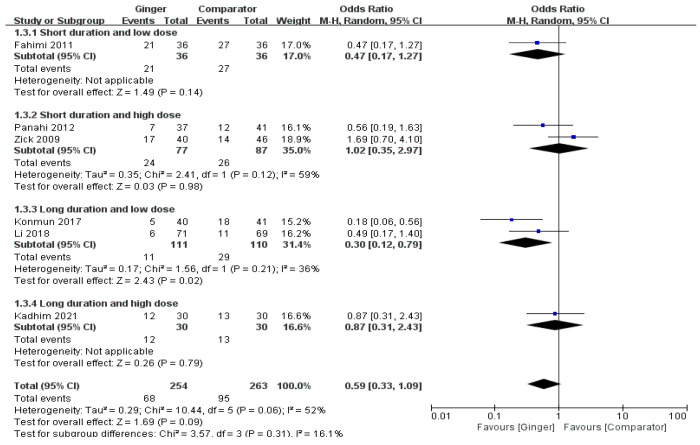
Effects of ginger intake on chemotherapy-induced acute vomiting. Short duration ≤ 4 days; Long duration > 4 days; Low dose ≤ 1 g/day; High dose > 1 g/day. Fahimi 2011 [19], Panahi 2012 [28], Li 2018 [23], Kadhim 2021 [21].

**Figure 5 nutrients-14-04982-f005:**
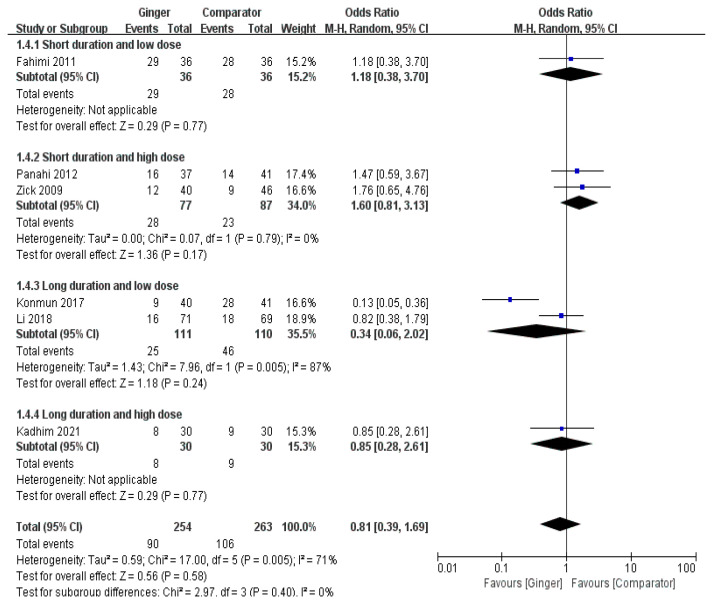
Effects of ginger intake on chemotherapy-induced delayed vomiting. Short duration ≤ 4 days; Long duration > 4 days; Low dose ≤ 1 g/day; High dose > 1 g/day. Fahimi 2011 [19], Panahi 2012 [28], Li 2018 [23], Kadhim 2021 [21].

**Table 1 nutrients-14-04982-t001:** Characteristics of the included RCT.

Author, Year [Reference]	Country	Study Design	Baseline Age (Years), Mean ± SD	Type of Cancer	Sample Size	CTx Emetogenicity	Intervention				Comparator	Outcome
							Intervention	Intervention Dose	Intervention Duration	Antiemetic Agent Use (Y/N)		
Alsparslan CB et al., 2012 [14]	Turkey	RCT	-	Hematological cancer	45	Y (NI)	Tablets	1600 mg/d (800 mg × 2 times)	(NI)	N	NI (Steron IV3 mg, NI)	Nausea and vomiting ↓ (*p* < 0.05)
Ansari, 2016 [15]	Iran	RCT	Range, 25–79; Mean, 48.56	Breast cancer	119	AC, CAF, or TAC	Capsules (Powdered dried ginger root)	1000 mg/d (500 mg × 2 times)	3 days	Y(Aprepitant 125 mg, Granisetron 3 mg IV)	Placebo (Aprepitant 125 mg, Granisetron 3 mg IV, Starch)	Nausea severity (NSD)Vomiting severity (NSD)
Arslan, 2015 [16]	Turkey	RCT	Range, 49–58; Mean, 48.5	Breast cancer	60	Anthracycline, Cyclophosphamide, Doxorubicin, 5-fluorouracil	Powder (Dried ginger root) + Yogurt	1000 mg/d (500 mg × 2 times)	3 days	Y (Dexamethasone, Ranitidine)	Placebo (Dexamethasone, Ranitidine, NI)	Nausea severity ↓ (*p* < 0.05) Number of vomiting ↓ (*p* < 0.05) Number of retching (NSD)
Bossi, 2017 [17]	Italy	RCT	Ginger: 58.8 ± 10.4, Placebo: 59.5 ± 10.0	Head and neck, Lung, Other	244	Cisplatin in combination with other agents such as 5-fluorouracil, Adryamicine, Epiadriamycine, etc.	Gelatin soft gel capsules (40 mg of standardized ginger CO₂ supercritical extract)	160 mg/d (80 mg × 2 times)	From 42 to 56 days	Y (NK-1 receptor antagonist, 5-HT3 receptor antagonist, Dexamethasone)	Placebo (NK-1 receptor antagonist, 5-HT3 receptor antagonist, Dexamethasone, Sunflower oil)	Nausea and vomiting (NSD)
Das, 2020 [18]	India	RCT	Range, 26–60	Not specified	100	Y (NI)	Ginger tea	NI	5 weeks	N	N	Nausea and vomiting↓ (*p* < 0.0001)
Fahimi, 2011 [19]	Iran	RCT	50.53 ± 12.21	Non-solid tumor, Solid tumor, Lung cancer, Others	36	Cisplatin in combination with other agents	Capsules (Powdered ginger)	1000 mg/d (500 mg × 2 times)	3 days (3 week washout)	Y (5-HT3 antagonist, Corticostreoid)	Placebo (5-HT3 antagonist, Corticostreoid, Lactose)	Prevalence of nausea and vomiting (NSD)Severity of nausea and vomiting (NSD)
Shokri, 2017 [20]	Iran	RCT	Ginger: 52.7 ± 10.55, Control: 52.69 ± 15.56	Ovarian cancer	49	Carboplatin, Paclitaxel	Ginger capsules	2 g/d	6 cycles	N	Placebo (N, NI)	Nausea and vomiting (NSD)
Kadhim, 2021 [21]	Iraq	RCT	≥ 21 y	Breast, Colorectal, Gynecological, Lung cancer, Others	60	Y (NI)	Ginger tea	1.5 g/day	5 days	Y (NI)	Y (NI)	Acute and delayed vomiting (NSD) Acute and delayed nausea (NSD) Severity of nausea ↓ (*p* < 0.05)
Konmun, 2017 [22]	Thailand	Double-blind placebo RCT	Range, 19–81; Median, 53	Breast, Ovarian cancer, Other	81	Anthracycline-based, Platinum-based, Other	Capsules(6-gingerol)	20 mg/day (10 mg × 2 times)	12 weeks	Y(Ondansetron, Metoclopramide, Dexamethasone)	Placebo(Ondansetron, Metoclopramide, Dexamethasone, Microcrystalline cellulose+ Colloidal silicon dioxide)	Vomiting ↓ (*p* < 0.001) Severity of nausea ↓ (*p* < 0.001)
Li X, 2018 [23]	China	Double-blind placebo RCT	Placebo: 57.46 ± 7.82; Ginger: 57.52 ±7.24	Lung cancer	140	Cisplatin based regimens	Capsules (Powdered dried ginger root)	0.5 g/day (0.25 g × 2 times)	5 days	Y (5-HT3 RAs)	Placebo (5-HT3 RAs, Corn starch)	Acute and delayed nausea (NSD), Acute and delayed vomiting (NSD)
Manusirivithaya, 2004 [24]	Thailand	Cross-over double-blind RCT	Regimen A→B: 46.8 ± 9.7; Regimen B→A: 46.2 ±14.4	Gynecologic cancer	43	Cisplatin-based chemotherapy	Capsules (Powdered dried ginger root)	1.0 g/day (0.25 g × 4 times)	5 days	Y (Metoclopramide)	Placebo (Metoclopramide, Corn starch)	Acute nausea, Delayed nausea, Acute vomiting, Delayed vomiting (NSD)
Marx W et al., 2017[25]	Australia	Double-blind placebo RCT	Mean ± SD 58 ± 12	Breast, Colon, Lymphoma, Other cancer	34	Y (NI)	Capsules (5% gingerols)	1.2 g/day (0.3 g × 4 times)	5 days	Y (Aprepitant)	Placebo (Aprepitant, NI)	Nausea and vomiting (NSD)
Montazeri, 2013 [26]	Iran	Cross-over double-blind RCT	50.3 ± 13.1	Cancer	31	Cisplatin or cisplatin in combination with fluorouracil 5	Capsules (Powdered ginger)	1.0 g/day (0.5 g × 2 times)	28 days (23 days washout period)	Y (Ketril, Dexametasone, Metocolopramid)	Placebo (Ketril, Dexametasone, Metocolopramid, Chickpea powder)	Frequency and intensity of nausea and vomiting↓(*p* value = not presented)
Muthia R et al., 2013 [27]	Indonesia	RCT	-	Breast cancer	20	Cyclophosphamide, Adriamycin-5-fluoro uracil	10% crude drugs (10 g of zingiber officanale varietas rubrum + 100 mL of water) (for 10 serving)	3 g/day (1 g × 3 times)	5 days	Y (Ondasentrom, Dexamethasone)	Control (Ondasentrom, Dexamethasone)	Nausea and vomiting↓ (*p* = 0.036)
Panahi, 2012 [28]	Iran	Open pilot RCT	Range 35–74; Mean ± SD 51.83 ±9.18	Breast cancer	78	Docetaxel, Epirubicin, Cyclophosphamide	Capsules (Powdered dried ginger root)	1.5 g/day (0.5 g × 3 times)	4 days	Y (Granisetron, Dexamethasone)	Control (Grainsetron, Dexamethasone)	Prevalence of acute nausea ↓ (*p* = 0.04) Prevalence of acute vomiting and retching (NSD) Prevalence of delayed nausea, vomiting and retching (NSD)
Ryan, 2012 [29]	USA	RCT	≥ 18 years; Mean 53	Any	371	Y (NI)	Capsules (Liquid extract of ginger root)	0.5 g/day (0.25 g × 2 times)	6 days	Y (5-HT3 receptor antagonist, Dexamethasone)	Placebo (5-HT3 receptor antagonist, Dexamethasone, Extra virgin olive oil)	Severity of acute nausea ↓ (*p* = 0.017) Severity of delayed nausea (NSD)
					375		Capsules (Liquid extract of ginger root)	1.0 g/day (0.5 g × 2 times)				Severity of acute nausea ↓ (*p* = 0.036) Severity of delayed nausea (NSD)
					375		Capsules (Liquid extract of ginger root)	1.5 g/day (0.75 g × 2 times)				Severity of acute nausea ↓ (*p* = 0.0001) Severity of delayed nausea (NSD)
Sontakk,2003 [30]	India	Cross-over RCT	Median, 47	Cancer	50	Cyclophosphamide	Ginger capsules (Ginger powder)	1000 mg/d	3 cycles	N	Placebo (Metoclopramide, Ondansetron, Lactulose)	Nausea and vomiting (NSD)
Sanaati, 2016 [31]	Iran	RCT	Range, 20–60	Breast cancer	30	Y(NI)	Capsules (Powdered ginger root)	1000 mg/d (500 mg × 2 times)	10 days	Y (Dexamethasone, Metoclopramide, Aprepitant)	Control (Dexamethasone, Metoclopramide, Aprepitant)	Number of nausea (*p* = 0.006) Number of vomiting (*p* < 0.0001)
Thamlikitkul, 2017[32]	Thailand	Cross-over RCT	Range, 32–68; Mean, 49	Breast cancer	34	Adriamycin, Cyclophospharmide	Ginger capsules (Dry ginger powder)	1000 mg/d (500 mg × 2 times)	5 days	Y (Ondansetron, Dexamethasone, Metoclopramide, Domperidone)	Placebo (Ondansetron, Dexamethasone, Metoclopramide, Domperidone, Inactive ingredients of the ginger)	Acute nausea and delayed nausea (NSD)
Uthaipaisanwong, 2020 [33]	Thailand	Cross-over RCT	53.9 ± 13.8	Ovarian, Cervical, Endometrial, Vulvar, cancer	47	Carboplatin-paclitaxel chemotherapy	Ginger capsules (Dried ginger)	2 g/d	5 days in each cycle (total 2 cycles)	Y (Dexamethasone, Ondansetron, Ranitidine, Dimenhydrinate IV)	Placebo (Dexamethasone, Ondansetron, Ranitidine, Dimenhydrinate IV, Corn starch)	Acute nausea ↓ (*p* = 0.03) Delayed nausea, Acute and delayed vomiting (NSD)
Wazqar, 2021 [34]	Egypt	RCT	Ginger: 36.8 ± 5.84, Control: 36.1 ± 5.22	Gynecological cancers	100	Cisplatin-based regimens	Ginger tea (Fresh green ginger root)	4 cups/d (1 cup = 250 mg of fresh green ginger roots to 100 mL of boiled water, simmer for 10 min + 1/2 teaspoon of honey for taste)	6 days	Y (Granisetron, Dexamethasone)	Control (Grainsetron, Dexamethasone)	Severity of nausea ↓ (*p* < 0.05)
Yekta, 2012 [35]	Iran	RCT	Ginger: 41.8 ± 8.4, Control: 45.1 ± 10	Breast cancer	80	Y (NI)	Ginger capsules (dry powdered ginger root)	1 g/d	6 days	Y (Kytril, Granisetron, Dexamethasone)	Placebo (Kytril, Granisetron, Dexamethasone, Starch)	Anticipatory, Acute and delayed vomiting ↓ (*p* = 0.04; *p* = 0.04; *p* = 0.003, respectively)
Zick S.M. et al., 2009 [36]	US	Placebo RCT	Mean ± SD 1 g ginger: 53.3 ± 12.0, 2 g ginger: 58.3 ± 12.3, control: 55.5 ± 11.2	Cancer	162	Cisplatin, Cyclophosphamide, Dacarbazine, etc.	Ginger capsules (dry extract of ginger root)	1 g/d (0.5 g × 2 times)	3 days	Y (5-HT3 receptor antagonist, Aprepitant)	Placebo(5-HT3 receptor antagonist, Aprepitant, Lactose)	Acute and delayed nausea (NSD) Acute and delayed vomiting (NSD)
							Ginger capsules (dry extract of ginger root)	2 g/d (1 g × 2 times)				Acute and delayed nausea (NSD) Acute and delayed vomiting (NSD)

NI, no information; NSD, no significant difference; AC, doxorubicin 60 mg/m^2^+cyclophosphamid 600 mg/m^2^; CAF, cyclophosphamide 500 mg/m^2^+doxorubicin 50 mg/m^2^ +5-Fluorouracil 500 mg/m^2^; TAC, docetaxel 75 mg/m^2^+doxorubicin 50 mg/m^2^+ cyclophosphamide 500mg/m^2^.

**Table 2 nutrients-14-04982-t002:** Risk of bias assessment for the included RCTs.

Author, Year [Reference]	Risk of Bias Assessment for the Included RCTs					
	Bias Arising from the Randomization Process	Bias Due to Deviations from the Intended Interventions	Bias Due to Missing Outcome Data	Bias in the Measurement of the Outcome	Bias in the Selection of the Reported Results	Overall Bias
Alsparslan CB et al., 2012 [14]	Some concerns	Some concerns	Low	Low	Low	Some concerns
Ansari, 2016 [15]	Low	Low	Low	Low	Low	Low
Arslan, 2015 [16]	High	Some concerns	Low	High	Low	High
Bossi, 2017 [17]	Low	Low	Low	Low	Low	Low
Das, 2020 [18]	Low	Some concerns	Low	High	Low	Some concerns
Fahimi, 2011 [19]	Low	Low	Low	Low	Low	Low
Shokri, 2017 [20]	Some concerns	Some concerns	Low	Some concerns	Low	Some concerns
Kadhim, 2021 [21]	Some concerns	High	Low	High	Low	Some concerns
Konmun, 2017 [22]	Low	Low	Low	Low	Low	Low
Li X, 2018 [23]	Low	Low	Low	Low	Low	Low
Manusirivithaya, 2004 [24]	Low	Low	Low	Low	Low	Low
Marx W et al., 2017 [25]	Low	Low	Low	Low	Low	Low
Montazeri, 2013 [26]	Low	Some concerns	High	Low	Low	Some concerns
Muthia R et al., 2013 [27]	Some concerns	Some concerns	Some concerns	Low	Low	Some concerns
Panahi, 2012[28]	High	High	Some concerns	Low	Low	Some concerns
Ryan, 2012 [29]	Low	Low	Low	Low	Low	Low
Sontakke, 2003 [30]	Low	Low	Low	Low	Low	Low
Sanaati, 2016 [31]	Low	High	Some concerns	High	Low	High
Thamlikitkul, 2017 [32]	Low	Some concerns	Low	Low	Some concerns	Some concerns
Uthaipaisanwong, 2020 [33]	Low	Low	Low	Low	Low	Low
Wazqar, 2021 [34]	Low	High	Low	High	Low	Some concerns
Yekta, 2012 [35]	Low	Low	Low	Low	Low	Low
Zick, 2009 [36]	Low	Low	Low	Low	Low	Low

**Table 3 nutrients-14-04982-t003:** Reported adverse events after intake of ginger.

Author, Year [Reference]	Adverse Events (Number of Events)
Alsparslan CB et al., 2012 [14]	No information was reported.
Ansari, 2016 [15]	No adverse events occurred during the study.
Arslan, 2015 [16]	No adverse events occurred during the study.
Bossi, 2017 [17]	Mild gastrointestinal events such as dyspepsia, abdominal, epigastric pain, hiccups (63 in the ginger group; 35 in the placebo group); Serious adverse events such as hospitalization, prolonged hospitalization (15 in the ginger group; 18 in the placebo group).
Das, 2020 [18]	No adverse events occurred during the study.
Fahimi, 2011 [19]	No information was reported.
Shokri, 2017 [20]	Adverse events without specific information (8 in the ginger group; 20 in the control group).
Kadhim, 2021 [21]	No information was reported.
Konmun, 2017 [22]	No adverse events occurred during the study; Significant lower fatigue when compared with placebo (2 in the ginger group).
Li X, 2018 [23]	No adverse events occurred during the study.
Manusirivithaya, 2004 [24]	No adverse events occurred during the study.
Marx W et al., 2017 [25]	Adverse events such as constipation and reflux (6 in the ginger group), but with no difference in adverse effects compared to placebo.
Montazeri, 2013 [26]	No information was reported.
Muthia R et al., 2013 [27]	No information was reported.
Panahi, 2012 [28]	No information was reported.
Ryan, 2012 [29]	Adverse events such as gastrointestinal symptoms, heartburn, bruising/flushing, rash (9 in the ginger group).
Sontakke, 2003 [30]	No adverse events occurred during the study.
Sanaati, 2016 [31]	No information was reported.
Thamlikitkul, 2017 [32]	Adverse events such as fever, fatigue, mucositis, diarrhea, constipation, neutropenia, thrombocytopenia, but with no difference in adverse effects compared to placebo.
Uthaipaisanwong, 2020 [33]	No serious adverse events occurred during the study.
Wazqar, 2021 [34]	No adverse events occurred during the study.
Yekta, 2012 [35]	The only reported adverse event during the study was heartburn, but with no difference in adverse effects compared to placebo.
Zick, 2009 [36]	Adverse events such as upper extremity deep vein thrombosis, severe diarrhea and abdominal pain, anemia, low platelets and white blood cells, but with no difference in adverse effects compared to placebo.

## Data Availability

The original contributions presented in the study are included in the article/Appendix A, further inquiries can be directed to the corresponding author/s.

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
