# Peer review of "Effects of Ginger Intake on Chemotherapy-Induced Nausea and Vomiting: A Systematic Review of Randomized Clinical Trials"

_nutrients, 2022, doi:10.3390/nu14234982_

Round 1
Reviewer 1 Report
Anyway, please check your manuscript throughout, because I found some typos.( 5-HT3 or 5-HT3)
Author Response
"Please see the attachment."

Reviewer 2 Report
On a quick search on PubMed, I found over 60 review articles related to ginger usage for nausea and vomiting chemotherapy-induced. The newest article was published in Sept 2022 in
Int J Mol Sci. 2022 Sep 24;23(19):11267. doi: 10.3390/ijms231911267. ( 5 studies and 4 reports included in the review ) and reviews the efficacy and safety in breast cancer patients going under chemotherapy.
The present article systematically reviewed 23 RCTs. ( row 29) The effects of ginger supplementation were compared to those of placebo or antiemetic agents. However, I am not clear if ginger is more effective in certain chemotherapeutic agents versus others. Also what group of patients would benefit the most? Is ginger effectiveness related to the type of tumor or chemotherapy? Or the review is too small for this type of conclusion. How the ginger should be taken for maximum effectiveness?
Table 1: Outcome does not have a consistent format: some capital letters were used, sometimes arrows – would help to add on all rows, not just say “change”. Needs more detail regarding the control. What was the standard at the time of RCT?
I would prefer to see a comparison between
1.RCTs placebo and ginger
2. RCT ginger and standard anti-emetic therapy
Is ginger significantly better for acute vomiting compared with standard or with placebo? Needs clarification.
What are the adverse events potentially related to ginger consumption and at which dose? Since high doses, may increase the risk of bleeding, this aspect should be mentioned.
The limitations of this review should be stated more clearly.
Author Response
"Please see the attachment."

Reviewer 3 Report
The author reported effects of ginger intake on chemotherapy-induced nausea and vomiting. The findings are potentially informative for the readers of nutrients But I have several minor concerns and I think that some additional information and rational explanation should be essential for readers. Minor comments 1. The type of chemotherapy is not standardized, but does it affect the effect of ginger? 2. Concomitant anti-nausea medications also differ from study to study, but does this also have an effect?Author Response
"Please see the attachment."
